# Genomic Profiling and Molecular Characterisation of Metastatic Urothelial Carcinoma

**DOI:** 10.3390/medicina60040585

**Published:** 2024-03-31

**Authors:** Gaetano Pezzicoli, Federica Ciciriello, Vittoria Musci, Silvia Minei, Antonello Biasi, Anna Ragno, Paola Cafforio, Mimma Rizzo

**Affiliations:** 1Department of Interdisciplinary Medicine, University of Bari “Aldo Moro”, 70124 Bari, Italy; gaetano.pezzicoli@uniba.it (G.P.); f.ciciriello3@studenti.uniba.it (F.C.); v.musci4@studenti.uniba.it (V.M.); s.minei@studenti.uniba.it (S.M.); a.biasi3@studenti.uniba.it (A.B.); paola.cafforio@uniba.it (P.C.); 2Medical Oncology Unit, Azienda Ospedaliera Universitaria Consorziale, Policlinico di Bari, 70124 Bari, Italy; anna.ragno@policlinico.ba.it

**Keywords:** urothelial carcinoma, FGFR, HER2, ADC, genomic profiling, molecular characterisation, biomarker, targeted therapy

## Abstract

The clinical management of metastatic urothelial carcinoma (mUC) is undergoing a major paradigm shift; the integration of immune checkpoint inhibitors (ICIs) and antibody–drug conjugates (ADCs) into the mUC therapeutic strategy has succeeded in improving platinum-based chemotherapy outcomes. Given the expanding therapeutic armamentarium, it is crucial to identify efficacy-predictive biomarkers that can guide an individual patient’s therapeutic strategy. We reviewed the literature data on mUC genomic alterations of clinical interest, discussing their prognostic and predictive role. In particular, we explored the role of the fibroblast growth factor receptor (FGFR) family, epidermal growth factor receptor 2 (HER2), mechanistic target of rapamycin (mTOR) axis, DNA repair genes, and microsatellite instability. Currently, based on the available clinical data, FGFR inhibitors and HER2-directed ADCs are effective therapeutic options for later lines of biomarker-driven mUC. However, emerging genomic data highlight the opportunity for earlier use and/or combination with other drugs of both FGFR inhibitors and HER2-directed ADCs and also reveal additional potential drug targets that could change mUC management.

## 1. Introduction

The incidence of urothelial carcinoma (UC) has increased in recent years. UC currently ranks tenth among the most common cancers worldwide [1]. The five-year specific survival varies significantly by disease stage at diagnosis: 90% for in situ disease, 70% for localised disease, 39% for locally advanced disease, and 8% for metastatic disease [2]. Despite diagnostic and therapeutic advances in recent years, advanced UC still has a poor prognosis [3].

For more than two decades, platinum-based chemotherapy has been the standard of care in neoadjuvant [4], adjuvant [5], and metastatic settings [6]. In recent years, immune checkpoint inhibitors (ICIs) have gained a crucial role in the UC therapeutic armamentarium. ICIs have been shown to improve disease-free survival in the adjuvant setting, and recent literature data support the benefit of ICIs in pathological response rate, event-free survival (EFS), and relapse-free survival (RFS) in the neoadjuvant setting [7]. In the first-line setting, the results obtained with an ICI in combination with platinum-based chemotherapy [8] or an antibody–drug conjugate (ADC) [9] as well as maintenance monotherapy for patients progression-free to first-line, platinum-based chemotherapy [10] are statistically significant and clinically relevant. Based on the recent results of the randomised phase III trial EV-302, which demonstrated a significant superiority of the experimental arm over platinum-based, first-line chemotherapy in both cisplatin-eligible and cisplatin-ineligible patients, the combination of enfortumab vedotin, an ADC, and pembrolizumab, an anti-PD-1 inhibitor, represents a new standard of care for previously untreated mUC patients [11]. Moreover, patients who do not receive ICI as a first-line or maintenance therapy benefit from second-line immunotherapy [12].

The results obtained with ADCs in subsequent lines are also remarkable: (I) Enfortumab vedotin (a fully human monoclonal antibody specific for nectin-4 linked to monomethyl auristatin E) was proved to be superior to chemotherapy in patients with mUC that progressed after platinum-based chemotherapy and ICI [9]; (II) Sacituzumab govitecan (a TROP-2-directed antibody–drug conjugate with SN-38) showed notable activity in patients affected by mUC that progressed after platinum-based chemotherapy and ICI in a phase II trial [13], and a phase III trial is ongoing; (III) Trastuzumab deruxtecan (an immunoconjugate composed by an antibody directed against HER2 linked to a topoisomerase I inhibitor) showed efficacy in HER2-expressing UC in a phase II basket trial [14].

However, despite the considerable increase in therapeutic options for mUC patients, there is still one major absentee: targeted therapy. In the last decade, the only positive phase III trial with a molecular targeted drug has been the THOR trial, which evaluated the efficacy of erdafitinib, an FGFR inhibitor, in mUC-pretreated patients [15]. Targeted therapy could greatly expand the mUC therapeutic armamentarium, as already observed for many other solid tumour types, and the development of biomarker-driven therapeutic strategies would offer patients a greater number of therapeutic combinations and sequences [16].

This paper aims to review the most recent literature in order to depict a clear landscape of molecular alterations in mUC and their potential role in guiding therapeutic choices.

## 2. Genomic Alterations of Clinical Interest in UC

The backbone of the current knowledge on UC genomic alterations was the analysis of The Cancer Genome Atlas (TCGA) cohort, published in 2014 [17]. Our current description focuses on genomic alterations that could harbour therapeutic implications, present and future, as described below. Data from the cited clinical trials are summarised in Table 1.

### 2.1. FGFR Alterations

The FGFR is a transmembrane tyrosine kinase receptor consisting of a split intracellular tyrosine kinase domain and three extracellular immunoglobulin-like domains [22]. The native ligands of the FGFR are eighteen secreted molecules, fibroblast growth factors (FGFs), which bind to their receptor (FGFR1-4) and induce dimerization that activates downstream signalling, causing activation of the RAS/MEK/ERK and PI3K/AKT signalling pathways [23,24,25]. The FGFRs are involved in cell survival, including proliferation, growth, and differentiation, and play an important role in angiogenesis [26,27]. The role of FGFR family receptors in cancerogenesis is better depicted in Figure 1.

Many studies have proven the involvement of the FGFR in different cancers, including leukaemia, breast, prostate, and urothelial cancer [28]. In the latter, and in particular, in low tract UC, the alterations of FGFR are associated with a better prognosis; it is more frequent in non-muscle invasive bladder cancer, in low-grade and low-stage tumours, and luminal papillary malignancies. Moreover, some studies reported mutations of FGFR3 being associated with a lower T cell infiltration and also a lower probability of progression toward a metastatic stage [29,30]. FGFR-altered urothelial carcinomas are prone to respond to targeted therapies with FGFR inhibitors [31]. In recent years, many phase II/III studies have described the activity of FGFR inhibitors in UC clinical trials.

The first molecule that showed clinical activity is erdafitinib, an FGFR1–3 inhibitor [32,33]. A phase II trial, open-label, described the clinical outcome of erdafitinib in 99 patients with FGFR2/3-altered advanced UC who progressed during or after platinum chemotherapy. The ORR was 40% with an mPFS of 5.5 months and an mOS of 13.8 months [34]. A phase II trial, open-label, enrolling 87 cisplatin-ineligible mUC patients, evaluated the activity of erdafinitib monotherapy vs. erdafitinib plus cetrelimab (an anti-PD-1 antibody) in a first-line setting. The results, recently published, showed a meaningful activity of the combination (ORR 54.5%) and also demonstrated a good activity of erdafitinib monotherapy in FGFR-altered, cisplatin-unfit mUC patients [35]. A randomised phase III study (THOR) evaluated patients with FGFR2/3-altered mUC who progressed after one or two prior treatments, including anti-PD-(L)1 agents and chemotherapy, by comparing erdafitinib vs. investigator’s choice chemotherapy. Erdafitinib showed superior efficacy (mOS 12.1 months vs. 7.8 months; mPFS: 5.6 months vs. 2.7 months) with a manageable safety profile [36].

Pemigatinib, an FGFR1–3 inhibitor, showed, in a phase I/II study (FIGHT-101), a meaningful safety, tolerability, and response rate in solid malignancies, including urothelial cancer, driven by FGFR fusions/rearrangements and mutations [37]. The promising data led to a phase II trial in UC (FIGHT-201) that demonstrated preliminary efficacy in previously treated mUC patients with FGFR3 alterations [38]. A phase II study of pemigatinib monotherapy vs. pemigatinib plus pembrolizumab (anti-PD-1 antibody) is currently ongoing in first-line settings for cisplatin-unfit patients [39].

Another strong and selective FGFR1–4 inhibitor, rogaratinib, showed antitumour activity in a preclinical model [40]. A phase II/III study (FORT-1) was performed, confirming its activity in UC patients with FGFR alterations, selected by evaluating FGFR1/3 mRNA expression [41]. The FORT-1 trial showed that rogaratinib has an efficacy comparable with standard chemotherapy and an acceptable safety profile [41]. A phase Ib/II study of rogaratinib monotherapy vs. rogaratinib plus atezolizumab (anti-PD-1 antibody) in a first-line setting for cisplatin-unfit patients is currently ongoing [42].

A selective oral FGFR inhibitor, infigratinib, has promising clinical activity and tolerability in advanced UC patients with FGFR3 alterations [18]. A double-blind, randomised, placebo-controlled, phase III trial (PROOF-302) was evaluating the adjuvant setting in patients with muscle-invasive UC with FGFR3 alterations. The trial was stopped early by the sponsor [43].

Futibatinib (TAS-120) is a highly selective FGFR1–4 inhibitor that showed a good safety profile in advanced solid tumours [44]. A phase II study is ongoing to evaluate the combination of futibatinib with pembrolizumab in advanced or metastatic UC [45].

Dovitinib is an FGFR1–3 inhibitor with minor efficacy on FGFR4, as demonstrated in a preclinical model of breast cancer cell lines [46]. A phase II trial of dovitinib in FGFR-3-mutated or wild-type patients with pretreated mUC did not show sufficient efficacy; the endpoint was not met, and the trial was stopped [47].

Vofatamab (B-701), an FGFR3-specific monoclonal antibody, was proved to be safe in a phase Ib/II study (FIERCE-21) in relapsed or refractory mUC in combination with docetaxel chemotherapy [48]. Another phase Ib/II study (FIERCE-22) is ongoing to evaluate the clinical activity of vofatamab in combination with pembrolizumab in FGFR wild-type, metastatic urothelial carcinoma [49].

Derazantinib (ARQ 087), an orally bioavailable, multi-kinase inhibitor with potent pan-FGFR activity, has shown preliminary therapeutic activity against FGFR2 fusion-positive, intrahepatic cholangiocarcinoma. Clinical trials need to explore this promising molecule for the treatment of UC [50].

Most of the aforementioned clinical trials include both patients with FGFR mutations and patients with FGFR fusions and prove the efficacy of FGFR inhibitors in the overall population. However, the different FGFR alterations do not have the same predictive value for response to FGFR inhibitors [51]. It has been demonstrated in animal models that FGFR inhibitors are most effective against FGFR3 R248C, among all point mutations examined [51]. In study NCT02365597 [34], the overall ORR of erdafitinib was 40%, but in the subgroup of patients with FGFR fusions, it was 16%. However, the ORR in the group of patients with FGFR3-TACC3 fusion, the most common of all FGFR fusions, was close to the general population (36%). In the THOR study, the ORR was comparable between the patients with FGFR fusions and those with FGFR mutations, but the patients with FGFR mutations had a better PFS [36]. The FIGHT-201 study provides a detailed description of the relationship between different FGFR alterations and radiological response to the FGFR1–3 inhibitor [38]. The ORR in the patients who received pemigatinib varied according to the FGFR3 alteration; in patients with the S249C, R248C, and G370C mutations, the ORR was 23%, 29%, and 29%, respectively, while it was only 9% in patients with the Y373C mutation and 17% in patients with FGFR3 fusions [38]. Therefore, pemigatinib is more effective for specific FGFR3 mutations than for FGFR3 fusions. It should, however, be noted that, in the major trials evaluating erdafitinib (both NCT02365597 and THOR) and pemigatinib (FIGHT-201), the principal FGFR alterations included are point mutations. FGFR fusions and rearrangements were only present in 13.7–25.3% of the FGFR-altered UC population. More data on the relationship between specific FGFR fusions and the response to FGFR inhibitors are needed to design a truly personalised treatment strategy.

The proper interpretation of genomic profiling could help to identify patients who are weakly responsive to FGFR-targeted therapies and predict early resistance. In addition, seriate tumour DNA detection could help to identify the acquisition of resistance mutations. The acquired mutations inducing resistance can occur both in the targeted oncogenic driver itself and in other pathways. In vitro studies proved that the main pathway implied in FGFR-inhibitor resistance is the PI3K-mTOR pathway [52]. Wang et al. demonstrated in vitro that the inhibition of PI3K can restore sensitivity to FGFR inhibitors in resistant cell lines [52]. Facchinetti et al. analysed the post-progression tumour DNA of UC patients with FGFR alterations treated with FGFR inhibitors and found that, in 38% of cases, resistance was related to the acquisition of one or more mutations in the FGFR tyrosine kinase domain. Interestingly, in 52% of cases, resistance was linked to the acquisition of new mutations in the PI3K-mTOR pathway [53].

### 2.2. HER2 Overexpression

The human epidermal growth factor receptor 2 (HER2) is a transmembrane tyrosine kinase receptor encoded by the ErbB2 gene, and it belongs to the ErbB protein family, more commonly known as the epidermal growth factor receptor family. HER2 regulates the activation of signalling pathways that promote cell growth and proliferation [54]. The amplification of the ErbB2 gene leads to overexpression and functional hyperactivation of HER2-dependant pathways, which promote carcinogenesis [55].

HER2 is altered in 20–30% of UC, and specifically, the amplification of HER2, detected by FISH, is a negative prognostic factor associated with lymphatic dissemination [56]. In the last few decades, its role in biomarker-driven treatments has been pivotal; research on HER2-directed agents in cancers with HER2 genomic alterations has led to the approval of new drugs [57].

The first therapy approved against HER2 is a humanised monoclonal antibody, trastuzumab, which binds to the extracellular domain of HER2, resulting in inhibition of the intracellular signal pathway [58]. In UC, trastuzumab was evaluated in a multicentre phase II trial in combination with chemotherapy (carboplatin, paclitaxel, and gemcitabine) in advanced UC. Fifty-seven (52.3%) of 109 registered patients were HER2/neu positive, and 70% of these patients had objective responses [59]. In a randomised phase II trial, the addition of trastuzumab to traditional chemotherapy did not improve clinical outcomes [60]. The HER2 basket trial MyPathway evaluated the double block with trastuzumab and pertuzumab in several cancers, including UC (22 patients). The reported ORR was only 15.8% [61].

Intracellular HER2 inhibitors act by limiting the intracellular signal of the HER2 receptor. They belong to the family of tyrosine kinase inhibitors. Lapatinib is the first intracellular HER2 inhibitor evaluated in breast cancer that showed an interesting activity in patients who progressed to a prior line with an anti-HER2 [62]. In a single-arm, phase II study with a combination of docetaxel and lapatinib in mUC, lapatinib did not show a meaningful activity and efficacy [63]. Similar results were reported in a phase III, double-blind, randomised trial that compared maintenance lapatinib versus placebo after first-line chemotherapy in mUC [64]. An irreversible pan-HER inhibitor, afatinib, demonstrated activity in biomarker-selected UC patients. In a phase II trial, afatinib showed a significant activity in patients with platinum-refractory UC with ERBB2 or ERBB3 alterations; a total of 83.3% with HER2 and/or ERBB3 alterations achieved a PFS of at least 3 months versus none of the patients without alterations [65]. However, a basket trial that evaluated afatinib in refractory solid tumours with HER2-activating mutations, including mUC, did not observe a meaningful response [66]. Another trial is ongoing to evaluate the activity of afatinib in UC with ERBB2/ERRB3 mutations, HER2 amplification, or EGFR amplification [67]. The promising pan-HER2 inhibitor drug neratinib showed, in a basket trial (SUMMIT), an interesting activity specifically in tumours that contained kinase-domain missense mutations, including UC [68].

In the last decade, a major breakthrough was the implementation of antibody–drug conjugates (ADC) in clinical practice. An ADC is a complex of a cytotoxic drug linked to a monoclonal antibody that is highly selective for a tumour antigen. The first one that was approved for clinical use was trastuzumab–emtansine (TDM-1), composed by an anti-HER2 monoclonal antibody (trastuzumab) linked to an anti-microtubule agent (DM-1) [69]. Although promising in preclinical activity, in a basket trial, TDM-1 did not show a response in UC patients [70]. A new multicentric, phase II clinical trial is ongoing to evaluate TDM-1 monotherapy or in combination with other drugs in UC [71]. Trastuzumab–deruxtecan (T-Dxd) is a new ADC, composed of an anti-HER2 antibody and a topoisomerase I inhibitor payload, that has significantly improved the prognosis of HER2-positive breast and gastric cancer [72,73]. Destiny-PanTumor02 is a global, open-label, multicentric, phase II trial evaluating the efficacy and safety of T-Dxd in patients with previously treated HER2-expressing solid tumours [14] (locally advanced, unresectable, or metastatic biliary tract, bladder, cervical, endometrial, ovarian, pancreatic, or other solid cancers, excluding breast, colorectal, gastric, and non-small-cell lung cancers). In all patients, the ORR was 37.1% (95% CI) with a 39.0% ORR in the bladder cohort. T-Dxd in combination with immunotherapy (nivolumab) was evaluated in a phase 1b trial in patients affected by HER2-positive UC and is a new therapeutic approach that showed promising antitumour activity; the ADC induces immunogenic cell death that can enhance the recruitment of immune cells in the tumour microenvironment, activating the immune response [74]. Two new ADCs, disitamab–vedotin (RC48-ADC) and trastuzumab–duocarmazine (SYD985), showed activity in two basket trials with a manageable safety profile in HER2-postive mUC patients who had failed at least one line of systemic chemotherapy [75,76].

Almost all the clinical trials mentioned in this section do not select their patients based on the presence of mutations but rather according to HER2 expression. Bellmunt et al. retrospectively analysed two cohorts of UC patients (*n* = 213) to highlight the relationship between HER2 overexpression by immunohistochemistry (IHC), fluorescence in situ hybridisation (FISH), DNA copy number, mRNA expression, and mutation status [77]. IHC and FISH results were almost always in concordance, and tumours with high HER2 expression also had higher levels of HER2 mRNA. Remarkably, HER2 gene mutations were detected in only 2% of patients and were HER2 L755S, which confers resistance to HER inhibitors, and HER2 V777L, which maintains sensitivity to HER inhibitors [78]. Assessments of other HER2-expressing tumours suggest other plausible mechanisms implied in HER2-targeting agent resistance: an inactive target receptor (such as HER2 C-terminal fragments—p95HER2 being resistant to trastuzumab) [79]; activation of target downstream components in other pathways, such as PI3K/Akt/mTOR and Ras-Raf-MAPK [80]; overexpression of other HER ligands or receptors [81]; and driver signalling from other receptors (such as the insulin-like growth factor-1 receptor (IGF1R)) [82].

### 2.3. Mutations of the PI3K/AKT/mTOR Axis

The PI3K/AKT/mTOR pathway is an intracellular signalling pathway that regulates the cell cycle, cell growth, differentiation, proliferation, autophagy, and cell movement. The alteration of this pathway, which can happen at multiple levels, has been linked to carcinogenesis, neo-angiogenesis, and therapy resistance [83]. The PI3K/Akt/mTOR signalling pathway is altered in a relevant proportion of urothelial cancers with a greater incidence in bladder tumours. The presence of PI3K/Akt/mTOR pathway-activating mutations correlates with a worse prognosis in UC patients [84].

The first drug targeting this pathway is the mTOR inhibitor everolimus. In a single-arm, phase II trial in 45 patients with metastatic UC progressing after one to four cytotoxic agents, everolimus did not meet its primary endpoint but demonstrated a minimal anti-tumour activity in a limited number of patients with advanced UC. Notably, none of these patients had mTOR alterations. These results were also confirmed in another phase II clinical trial [19,20].

Everolimus in combination with pazopanib, a tyrosine kinase inhibitor with anti-angiogenic activity, has been evaluated in a phase I trial in 19 mUC patients previously treated with one to three lines of chemotherapy [85]. Currently, new PI3K/mTOR inhibitor drugs are being evaluated in several clinical trials that also include patients with advanced UC.

Buparlisib, an oral pan-class I PI3K inhibitor that inhibits wild-type and altered PI3K, shows a modest activity in a phase II trial in platinum-refractory patients with mUC [86].

Eganelisib is the first highly selective PI3Kγ inhibitor, orally administered, with an antitumour activity demonstrated, in monotherapy or in combination with an anti-PD-1, in several solid tumours in a phase 1 trial. The MARIO-275 phase II, multicentric, randomised trial is ongoing, evaluating patients with advanced UC (NCT03980041) [87].

Sapanisertib, a potent inhibitor of mTOR complex 1 and 2, was evaluated in a phase II trial in patients with metastatic UC harbouring a TSC1 or TSC2 mutation; however, the trial was terminated early for futility with no response in 13 patients tested [88]. Despite these negative results, the biological rationale for the use of sapanisertinib in UC prompted the design of a phase II, multicentre, single-arm, open-label study of paclitaxel and sapanisertinb in mUC (NCT03745911).

Dactolisib, an oral pan-class I dual PI3K and mTOR complex 1/2 inhibitor, was evaluated in a multicentre phase II trial in patients with a locally advanced or metastatic transitional cell carcinoma (TCC) after failure of platinum-based therapy. Dactolisib showed modest clinical activity and an unfavourable toxicity profile in pretreated patients [21].

The association between immunotherapy and mTOR inhibitors was tested in a multi-arm, phase Ib study, the BISCAY trial, that included 29 pretreated mUC patients who received durvalumab plus an mTOR inhibitor, vistusertib (arm E). The combination treatment resulted in an ORR of 21% with 38% of patients discontinuing treatment due to adverse events [89].

### 2.4. DNA Damage Response Gene Alterations

The DNA damage response (DDR) is a critical element in cell homeostasis; when this mechanism is altered, genomic instability occurs, resulting in the accumulation of DNA mutations that induce cancer development [90]. Among the DDR genes, specifically in the homologous recombination repair (HRR) system, BRCA and PARP are two of the main genes involved. In BRCA-mutated cells, the deactivation of PARP causes a drastic drop in the efficacy of DNA repair with a consequent sharp increase in the frequency of DNA mutations, including death-inducing mutations [91]. The alteration of these genes is correlated with a worse prognosis in advanced UC [92]. On the other hand, DDR mutations are a potential target for therapy with PARP inhibitors in mUC patients [93].

As in other malignancies, in metastatic or advanced UC, PARP inhibitors are under investigation for their promising therapeutic approach, comprising the correlation with response to the immune checkpoint blockade [94].

The first poly ADP ribose polymerase (PARP) inhibitor, olaparib, improves survival in advanced ovarian, pancreatic, prostate, and breast carcinoma with DDR gene alterations and is used in clinical practice [95,96,97,98].

In mUC, the exact clinical role of PARP inhibitors is still undefined. A phase II study is ongoing to evaluate olaparib in monotherapy in mUC with DDR gene defects [99]. A phase II trial is ongoing to evaluate niraparib as maintenance in patients unselected for DDR mutational status who obtained disease control with first-line, platinum-based chemotherapy (NCT03945084). A concluded phase II, open-label trial did not show significant activity of rucaparib in previously treated mUC patients; however, the population was not selected for DDR mutation status [100].

Interestingly, alterations in DDR genes seem to predict the clinical response to ICI in mUC patients [101]. This harboured the hypothesis of the beneficial combination of ICI and PARP inhibitors. A phase Ib study, the BISCAY trial, is an ongoing study with an arm evaluating durvalumab plus olaparib in platinum-refractory, immuno-therapy naïve mUC patients with mutations in BRCA, ATM, or HRR genes. An ORR of 35.7% was observed [89]. Subsequent phase 2 studies were designed to evaluate the association between PARP inhibitors and ICIs, such as niraparib plus atezolizumab (NCT03869190) and durvalumab plus olaparib (NCT03459846: BAYOU trial, NCT03534492: NEODURVARIB trial).

Additionally, rucaparib is being evaluated in combination with lucitanib (a multi-tyrosine kinase inhibitor that showed efficacy on VEGFR1-2-3, FGFR1-2, and PDGFRα-β) or Sacituzumab–govitecan (an ADC composed of a humanised anti-trop-2 monoclonal antibody linked to the active metabolite of irinotecan, SN-38) in a phase Ib-II trial (NCT03992131).

### 2.5. Microsatellite Instability

Even though microsatellite instability (MSI) is not a proper target, it is undoubtedly of moderate importance in the clinical management of solid neoplasms. In 2017, MSI-H was declared as the first tissue/site-agnostic indication for the use of pembrolizumab (an anti-PD1 antibody) based on the result of five single-arm, multicohort, multicentre trials (KEYNOTE-016, -164, -012, -028, and -158). Tumours with MSI-H are well known for their good response to immunotherapy due to their hypermutated status, which leads to an increased neoantigen production that triggers effective immune responses [102]. MSI also plays a role as an ICI response predictor in mUC. Ma et al. reported a case of an MSI-H patient with mUC that experienced a sustained and prolonged response to ICI (sintilimab, an anti-PD-1 antibody) [103]. Sarfaty et al. reported a retrospective observation of 1333 mUC patients, including 26 MSI-H. These patients experienced deep and durable responses to ICI and poor responses to platinum-based chemotherapy [104]. Even more recently, a case was reported of a patient with MSI-H metastatic upper-tract UC (mUTUC) that had little to no response with chemotherapy but experienced a durable complete response with avelumab (an anti-PD-L1 antibody) and metastases-directed radiotherapy [105].

The role of MSI-H as a predictor of the ICI response in mUC should be taken into consideration when designing a strategy for mUC patients, since MSI-H is a feature present in 6% of the whole population and in even more mUTUC patients (9%) [106].

### 2.6. Other Targets of Potential Clinical Interest in UC

The research on the ALK alteration pathway was performed in different tumours including UC. However, Bellmunt J. et al. described a low prevalence of ALK gene alterations in mUC [107]. Two studies are ongoing to evaluate the efficacy of anti-ALK (crizotinib and erlotinib) in urothelial cancer (NCT02612194, NCT02091141).

Overexpression of the tyrosine kinase c-Met and its involvement with AKT/GSK signalling were found in a preclinical model of UC [108]. c-Met is directly activated by a cellular matrix kinase, SRC. A phase II trial in muscle-invasive urothelial carcinoma of the bladder evaluated dasatinib, an oral SRC-family kinase (SFK) inhibitor, in a neoadjuvant setting, showing limited clinical activity [109].

Bruton’s tyrosine kinase (BTK) has been shown to be a crucial agent in the progression of numerous solid tumour types including UC [110]. An increased level of BTK expression is associated with a phenotype of bladder cancer cells with a higher tendency toward invasion and metastasis [111]. A phase Ib/II study of ibrutinib (BTK inhibitor) and weekly paclitaxel in 29 patients as a second/third line of treatment showed 26% ORR [112].

## 3. Discussion

Despite remarkable diagnostic and therapeutic advances over the last decade, mUC remains associated with a dismal prognosis. The turning point could be the development of biomarker-driven, personalised approaches for mUC patients. Therapy personalisation has been a major game changer in modern oncology for many malignancies; a total of 50% of non-small-cell lung cancer patients with druggable oncogenic drivers [113], 50% of melanoma patients with druggable BRAFv600 mutations [114], 15–20% of breast cancer patients with HER2 positivity, and 10–15% of ovarian cancer patients with druggable DDR mutations benefit from specific targeted treatments [115].

In this review, we focused on the most frequent genomic alterations in UC and on biomarker-driven therapies for mUC supported by solid clinical evidence.

The data from large genomic profiling cohorts revealed that mUC harbours many druggable alterations. In the TCGA cohort [16], 34% of patients had FGFR1–3 alterations (17% mutations, 17% copy number variations), 14.6% of patients had BRCA1 or 2 mutations, 4.9% had ALK alterations (1.2% amplifications, 3.2% mutations, 0.002% fusions), 4.9% had ROS1 mutations, 6.6% had NTRK1 alterations (4.9% amplification, 1.7% mutations), 2.9% had NTRK2 alterations (1% amplifications, 1.9% mutations), and 1.5% had NTRK3 mutations.

FGFR is the most promising target in the mUC’s current therapeutic scenario. FGFR1, 2, and 3 can exert their oncogenic action as a consequence of a single-nucleotide mutation or an amplification. FGFR inhibitors act by binding to this receptor and shutting down its signal in both FGFR-mutated and FGFR-amplified UCs [24]. Many inhibitors targeting FGFR alterations—namely erdafitinib [36], pemigatinib [38], rogaratinib [41], infigratinib [43], and futibatinib [45]—showed efficacy in mUC patients. However, to date, the only drug that has passed all clinical trial phases is erdafitinib, an FGFR1–3 inhibitor [32,33] whose efficacy has been shown to be superior to the standard of care in patients with susceptible FGFR2/3 alterations who have failed previous chemotherapy and immunotherapy. [15] FGFR inhibitors are being tested in early disease settings—such as the THOR-2 phase II trial with erdafitinib in patients with NMIBC [116] or the PROOF-302 phase III trial with infigratinib as an adjuvant treatment [117]—and their results could help raise awareness of the importance of genomic profiling in localised UC. Moreover, recent preclinical data confirm that FGFR inhibition in FGFR3-mutant urothelial carcinoma abolishes immunosuppression and increases the efficacy of anti-PD-1 agents [118]. This synergy of action provides a strong rationale for investigating FGFR inhibitors in combination with ICIs for the first-line treatment of FGFR-altered mUC. The Norse phase II trial already showed an increase in efficacy in the first-line setting for the combination of cetrelimab (an anti-PD1 antibody) plus erdafitinib over erdafitinib monotherapy [35]. Further information will be acquired upon the completed recruitment of three other ongoing trials: the FORT-2 phase II trial (investigating the combination of rogaratinib and atezolizumab versus rogaratinib alone in FGFR1/3-altered, treatment-naïve mUC), the FIGHT-205 phase II trial (investigating the combination of pemigatinib and pembrolizumab versus pemigatinib alone versus chemotherapy in FGFR3-altered, treatment-naïve mUC), and the NCT04601857 phase II trial (investigating the combination of futibatinib and pembrolizumab in FGFR1/4-altered, treatment-naïve, cisplatin-ineligible mUC).

Another promising target could be HER2 due to the recent results of HER2-targeting ADC in mUC patients. The DESTINY-PanTumor02 trial demonstrated the efficacy of trastuzumab–deruxtecan in previously treated mUC patients with HER2 overexpression [14]. This evidence is of uttermost importance and quite unexpected, since neither HER2-targeted antibodies (such as trastuzumab [59]) nor HER2-targeted intracellular inhibitors (such as lapatinib [64] or afatinib [67]) showed encouraging results in previous trials. These results shed light on the true role of HER2 in UC biology, more to be intended as a bystander rather than a central driver. In fact, it should be noted that, among the patients with high HER2 expression, only 35% show amplification of the HER2 gene [119]. Therefore, in many cases, HER2 overexpression is the consequence of some other intracellular event that does not involve the HER2 gene. Nevertheless, given the recent results of trastuzumab–deruxtecan in HER2-positive mUC, in the next few years, HER2-expression assays could become a requirement of the mUC management algorithm. The next frontier for HER2-targeting ADC in mUC is the combination with ICI. The mechanisms of ICI and ADC synergy still need clarifications, but the efficacy of this combination has already shown promising results in the DS8201-A-U105 phase I trial (a study evaluating the combination of trastuzumab–deruxtecan and nivolumab in pretreated, biomarker-selected mUC patients) [71] and in the NCT03523572 phase II trial (a study evaluating the combination of disitamab–vedotin and toripalimab in pretreated, non-biomarker-selected mUC patients) [120].

The results are more modest for PI3K/Akt/mTOR-targeting agents (such as everolimus [19], sapanisetrib [88], or buparlisib [86]) whose role needs further studies to be clarified, both in terms of efficacy and patient selection and biomarker identification. Finally, DNA repair gene mutations could be a reliable biomarker for treatment with PARP-inhibitor agents, alone or in combination with ICI, but major trials exploring these drugs in mUC are still ongoing; particular attention should be given to niraparib plus atezolizumab (NCT03869190) and durvalumab plus olaparib (NCT03459846: BAYOU trial, NCT03534492: NEODURVARIB trial).

The data reported in our review support the importance of early and extensive molecular profiling in mUC to identify biomarkers for targeted treatments. However, it should be emphasised that not all the alterations are equal in terms of treatment response. This has major implications; the physician should not only require appropriate genomic profiling for UC patients but must also be able to correctly understand and contextualise the results. This is important for FGFR, whose mutations have a different impact on therapy response, but may be even more relevant for HER2; while many clinical results are reported in patients with high HER IHC expression, some HER2 mutations may render treatment useless, regardless of expression status [78]. Furthermore, genomic profiling performed repeatedly could also allow for the early detection of resistance mechanisms within the molecular target itself or parallel pathways, such as PI3K-mTOR mutations that confer resistance to FGFR inhibitors [53] and HER2-targeting agents [80].

Targeted treatments revolutionised the first-line therapy of several solid neoplasms, but in mUC, their efficacy seems underwhelming; to date, their use in metastatic first-line and localised disease is not supported by evidence. FGFR inhibitors and HER2-directed ADCs as monotherapy have gained a role in the mUC treatment sequence exclusively in later treatment lines. ICI and target treatment combination could change this attitude; should the ongoing combination randomised trials confirm the promising results reported in the early phase clinical trials, the mUC treatment algorithm would change further. The need to couple target therapies with ICI to obtain better efficacy results highlights the differences between mUC and other solid tumours, such as NSCLC, whose management relies on targeted monotherapies. This phenomenon has two possible interpretations: (I) many mutations identified as biomarkers in mUC are not strong drivers, and therefore, their inhibition would not be enough to stop tumour growth and spread; (II) the immune system dysregulation in mUC plays a much more important role than in other neoplasms, and therefore, the inclusion of ICI in a combination treatment will always improve the outcomes.

However, to prove these two statements by means of genomic profiling alone is not feasible because (I) genomic-only profiling gives little to no information on the status of the anti-cancer immune response; (II) the role of a mutation in the global cancer cell survival economy cannot be inferred from genomic-only profiling; and (III) genomic alterations are subject to variations in space (intra-patient genomic heterogeneity between the primary tumour and the metastases [121]) and time (mutational shift due to tumour progression and adaptation to cancer treatments [122]). HER2 is a perfect example of these latest concepts; it is frequently overexpressed in mUC, but the drugs that inhibit its function produce poor results. This could be explained by the fact that mUC cells rely more on other drivers, and that HER2 overexpression is a collateral phenomenon, which has a limited impact on the survival and proliferation of mUC cells. Therefore, a drug targeting one genomic alteration may elicit little effect if that alteration does not play a pivotal role in cancer cell survival. Conversely, FGFR alterations are well known to be pivotal drivers in mUC development [30], but their inhibition becomes fully exploitable only in later treatment lines [15], unless immunotherapy is added. As a consequence, in mUC, the role of the immune system cannot be overlooked, even in the presence of a strong driver.

To bypass this problem, a wider perspective is needed; if genomic profiling is not enough, we could look at transcriptomic data to gain more insight into the tumour mechanisms suitable for treatment. In fact, the transcriptome can help decipher the true configuration of the inner network of cancer cells much more than the simple knowledge of mutated genes; by observing the shift in gene expression, the true weight of each mutation can be inferred. In addition, transcriptomic analysis can identify another class of oncogenic drivers: those that depend not on a DNA mutation (and are, therefore, invisible to genomic profiling) but only on under- or overexpression [123].

Transcriptomic analysis also offers a view on another possibly more important landscape: the role of the immune system in mUC. Bulk RNA sequencing data from tumour cells, stromal cells, and immune cells can be interpreted through deconvolution algorithms to obtain insight into the cell populations infiltrating the tumour, their activity status, and their mutual interactions. Understanding tumour immune infiltration is the basis for understanding if immunotherapy could work in a certain patient and what priming could be needed to make it work better. In other words, gene-expression profiling data could help unravel the role of the immune system in mUC to better clarify the correct setting for targeted treatments, ICI, and their combinations.

## 4. Conclusions

Genomic profiling in mUC offers significant opportunities for effective personalised treatments. Currently, the genomic findings on actionable oncogenes in mUC have limited applicability to real-world clinical practice, but several investigations are underway to broaden this perspective. mUC could benefit from more studies focusing on transcriptomic analysis, which will integrate gene mutation data and gene expression data. This approach will attribute to each mutation the correct role in the cancer cell survival economy and inform on the immune system status, thus enabling an effective treatment personalisation.

## Figures and Tables

**Figure 1 medicina-60-00585-f001:**
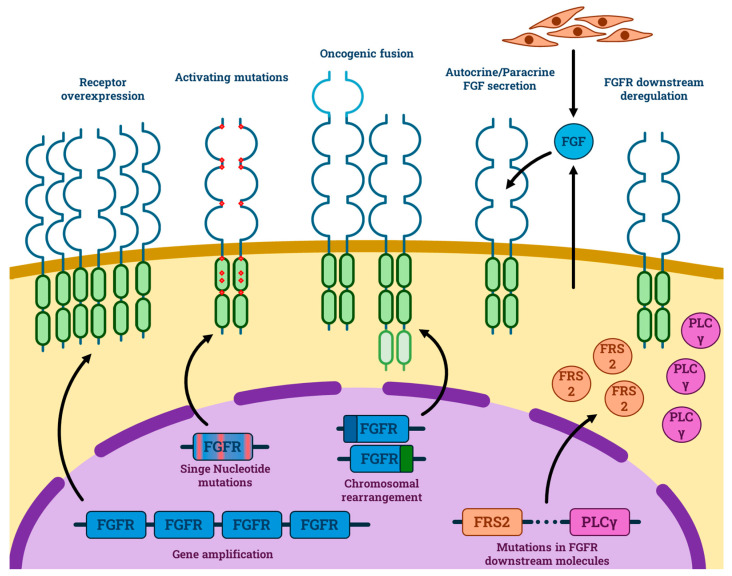
FGFR pathway dysregulation leading to cancerogenesis. The FGF receptor could be hyperactivated due to different events: receptor overexpression due to gene amplification; activating single nucleotide mutations (red spots); creation of constitutively active fusion proteins from chromosomal rearrangement (light green and light blue parts of the receptor); autocrine or paracrine FGF secretion; dysregulation of the downstream elements.

**Table 1 medicina-60-00585-t001:** Principal clinical trials of targeted therapy in mUC.

Target	Trial Name	Phase	Patients	Study Arms	Outcomes
FGFR alterations	NCT02365597	II	99 mUC pts, pretreated with CT and/or ICI, with FGFR3 point mutations or FGFR2/3 fusions	Single arm: Erdafitinib monotherapy	ORR: 40%; mPFS: 5.5 mts; mOS: 13.8 mts
	Norse trial (NCT03473743)	II	87 mUC pts treatment-naïve, cisplatin-unfit, with FGFR alterations	Erdafitinib + Cetrelimab (*n* = 44) vs. Erdafitinib (*n* = 43)	ORR: 54.5% vs. 44.2%; mPFS: 10.97 mts vs. 5.62 mts
	THOR trial (NCT03390504)	III	266 mUC pts pretreated with CT and/or ICI, with FGFR3 point mutations (R248C, S249C, G370C, or Y373C) FGFR2/3 fusions (FGFR2-BICC1, FGFR2-CASP7, FGFR3-TACC3, or FGFR3-BAIAP2L1)	Erdafitinib (*n* = 136) vs. investigator’s choice CT (*n* = 130)	mOS: 12.1 mts vs. 7.8 mts; ORR: 45.6% vs. 11.5%; mPFS: 5.6 mts vs. 2.7 mts
	FIGHT-201 (NCT02872714)	II	260 mUC pts pretreated with CT and/or ICI, with FGFR3 point mutations or fusions (Cohort A) or other FGF/FGFR alterations (Cohort B)	Pemigatinib (continuous or intermittent)	ORR: 24%; mPFS: 4 mts; mOS: 7 mts
	FIGHT-205 (NCT04003610)	II	mUC pts, treatment-naïve, with FGFR3 point mutations or rearrangements	Pembrolizumab + Pemigatinib vs. Pemigatinib vs. investigator’s choice CT	Ongoing
	FORT-1 (NCT03410693)	II/III	175 mUC pts pretreated with CT, with FGFR1/3 mRNA positive tumours	Rogaratinib (*n* = 88) vs. investigator’s choice CT (*n* = 87)	ORR: 20.7% vs. 19.3%; mPFS: 2.7 mts vs. 3.2 mts; mOS: 8.3 mts vs. 9.8 mts
	FORT-2 (NCT03473756)	Ib/II	27 mUC pts, treatment-naïve, with FGFR1/3 amRNA positive tumours	Rogaratinib + Atezolizumab vs. Rogaratinib	Phase II ongoing
	Pal et al. [18]	Ib/II	67 mUC pts, pretreated with CT and/or ICI, platinum-unfit, with FGFR3 point mutation or fusions	Single arm: Infigratinib	ORR: 25.4%; mPFS: 3.75 mts; mOS: 7.75 mts
	NCT04601857	II	mUC pts, treatment-naïve, cisplatin-ineligible, with or without FGFR3 point mutations or FGFR1-4 fusions	Single arm: Futibatinib + Pembrolizumab	Ongoing
	NCT00790426	II	44 mUC pts, pretreated with CT and/or ICI, with or without FGFR3 point mutations	Single arm: Dovitinib	ORR: 3.2%; mPFS: 3 mts
	FIERCE-21 (NCT02401542)	Ib/II	55 mUC pts, pretreated with CT and/or ICI, with or without FGFR3 point mutations or fusions	Vofatamab monotherapy or Vofatamab + Docetaxel	ORR: 12%
	FIERCE-22 (NCT02401542)	Ib/II	35 mUC pts, pretreated with CT, with or without FGFR3 point mutations or fusions	Vofatamab + Pembrolizumab	ORR: 30%
HER2 over- expression	NCT00151034	II	109 mUC pts, treatment naïve, expressing HER2	Single arm: Trastuzumab + Paclitaxel + Carboplatin + Gemcitabine	mPFS: 9.3 mts; mOS: 14.1 mts; ORR: 70%
	NCT01828736	II	61 mUC pts, treatment naïve, expressing HER2	Gemcitabine + Platin vs. Gemcitabine + Platin + Trastuzumab	mPFS: 10.2 mts vs. 8.2 mts; mOS: 15.7 mts vs. 14.1 mts; ORR: 65.5% vs. 53.2%
	NCT00949455	II	59 mUC pts, pretreated with platinum-based CT, with or without HER2 positivity	Single arm: Lapatinib	mOS: (HER2+ vs. HER2-) 7.5 mts vs. 2.5 mts
	NCT01382706	III	Maintenance after 1st line CT therapy for 232 mUC pts, expressing HER2	Lapatinib vs. Placebo	mPFS: 4.5 mts vs. 5.1 mts; mOS: 12.6 mts vs. 12.0 mts
	NCT02122172	II	15 mUC pts, previously treated with CT	Single arm: Lapatinib + Docetaxel	mOS: 6.3 mts; mPFS: 2.0 mts; ORR: 8%
	NCT02780687	II	25 mUC pts, previously treated with CT	Single arm: Afatinib	mPFS (HER2+ vs. HER2-): 6.6 mts vs. 1.4 mts
	LUX Bladder 1 trial (NCT01953926)	II	42 mUC pts, previously treated with CT, EGFR/HER2/ERBB3/4 mutated	Single arm: Afatinib	HER2+ cohort: mPFS 2.5 mts; mOS: 30.1 mts; ORR: 5.9%
	NCT04264936	II	19 mUC pts, previously treated with CT, without HER2 positivity	Single arm: Disitamab Vedotin	ORR: 26.3%; mPFS: 5.5 mts; mOS: 16.4 mts
	NCT03523572	II	32 mUC pts, previously treated with CT, with or without HER2 positivity	Single arm: Disitamab Vedotin + Toripalimab	ORR: 75%
	NCT04482309	I	30 mUC pts, previously treated with CT, with HER2 positivity	Trastuzumab Deruxtecan + Nivolumab	ORR: 36.7%; mPFS: 6.9 mts; mOS: 11.0 mts
Mutations of the PI3K/Akt/mTOR axis	Milowsky et al. [19]	II	45 mUC pts, pretreated with platinum-based CT	Single arm: Everolimus	ORR: 5%; mPFS: 2.6 mts; mOS: 8.3 mts
	Seront et al. [20]	II	37 mUC pts, pretreated with platinum-based CT	Single arm: Everolimus	ORR: 5.4%; mPFS: 2 mts; mOS: 3.3 mts
	NCT01184326	Ib/II	19 mUC pts, pretreated with platinum-based CT	Single arm: Everolimus + Pazopanib	ORR: 21%; mPFS: 3.6 mts; mOS: 9.1 mts
	NCT01551030	II	13 mUC pts, pretreated with platinum-based CT	Single arm: Burpalisib	ORR: 7.6%; mPFS: 3.2 mts; mOS: 8.7 mts
	NCT03047213	II	17 mUC pts, pretreated with platinum-based CT, with TSC1/2 mutations	Single arm: Sapanisertib	ORR: 0%; mPFS: NA; mOS: 3.4 mts
	Seront et al. [21]	II	20 mUC pts, pretreated with platinum-based CT	Single arm: BEZ235	ORR: 5%; mPFS: 2 mts; mOS: 4 mts
	BISCAY trial arm E (NCT02546661)	Ib	29 mUC pts, pretreated with platinum-based CT, with mTOR/PI3K alterations	Arm E: Vistusertib + Durvalumab	ORR: 24.1%; 6mts-PFS: 31.3%; 12mts-OS: 49%
DDR Gene alterations	NCT03375307	II	mUC pts with DDR gene alterations	Single arm: Olaparib	Ongoing
	ATLAS trial (NCT03397394)	II	97 mUC pts, pretreated with CT and/or ICI, with or without HRD-positivity	Single arm: Rucaparib	ORR: 0%; mPFS: 1.8 mts
	BISCAY trial arm C (NCT02546661)	Ib	14 mUC pts, pretreated with platinum-based CT, with DDR gene alterations	Arm E: Olaparib + Durvalumab	ORR: 35.7%; 6mts-PFS: 42.9%; 12mts-OS: 49.4%
	BAYOU trial (NCT03459846)	II	154 mUC pts, treatment-naïve, platinum-ineligible	Durvalumab + Olaparib vs. Durvalumab + Placebo	ORR: 28.2% vs. 18.4%; mPFS: 4.2 mts vs. 3.5 mts; mOS: 10.2 mts vs. 10.7 mts

Abbreviations: pts: patients; CT: chemotherapy; ICI: immune checkpoint inhibitors; ORR: objective response rate; mPFS: median progression-free survival; mOS: median overall survival; mts: months; mDoR: median duration of response.

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
