# Peer review of "Genomic Profiling and Molecular Characterisation of Metastatic Urothelial Carcinoma"

_medicina, 2024, doi:10.3390/medicina60040585_

Round 1

Reviewer 1 Report

Comments and Suggestions for Authors

The authors summarize in the manuscript  results from clinical trials to advocate the use of targeted therapy in the treatment of bladder cancer. 

The review is very nicely written but it is missing a deeper look into the correlation of genetic alterations and a successful response to drugs. That patients only respond to targeted therapies when showing alterations in the molecular target has been reviews already several times. The interesting question is why in for example patients with activating mutations in FGFR still only a subfraction responds to therapy. Question is, what kind of concomitant mutations have been found to date, what is known about activity of downstream targets and even more basic how many mutations in those molecules found in cancer have been really characterized as being functionally active or non-active.

The authors speculate in the discussion about combination of genomics with transcriptomics which might be beneficial but maybe before making the world even more complex, it should be discussed if we do really understand genetic mutations on a level that is sufficient to work with to date.

Thus, I would suggest to include to selected targets a detailed overview of mutations linked to activity (as for FGFR there are many very well performed studies available) and which of these mutations have been linked to therapy response. Then, it would be necessary to know, which of those mutations are really tested for in the trials. Only selected ones or complete sequencing. Also, correlation of FGFR activation and additional activation of pathways such as the PI3K signaling pathway with patient response would be interesting.  

Author Response

Thank you for your valuable suggestions, which we welcomed and incorporated into the manuscript. We implemented Table 1 to more accurately report the mutations allowed in each clinical trial. In addition, we implemented the section 'Genomic alterations of clinical interest in UC' with an overview of mutations associated with activity and therapeutic response. Furthermore, we discussed the occurrence of resistance based on acquired mutations, including mutations in the PI3K-mTOR axis.

Reviewer 2 Report

Comments and Suggestions for Authors

Author Response

  1. Please find one or two references for Dovitinib, Futibatinib, Vofatamab, and Derazanib on page 6.

> Reference from 42 to 47 concern Dovitinib, Futibatinib, Vofatamab, and Derazanib and are listed on page 6.

  1. The HER2 pathway is graciously illustrated diagrammatically by the authors.

> Thank you for your appreciation.

  1. On page 8, the authors kindly include one or two references for Buparlisib, Eganelisib, Sapanisertib, and Dactolisib.

> Reference from 85 to 88 concern Buparlisib, Eganelisib, Sapanisertib, and Dactolisib and are listed on page 8.

  1. Are there limitations on the use of genomic profiling?

> The main limitation of genomic profiling is the poor insight into the functional status of each mutation. This limitation could be overcome by using transcriptomic data, as reported in the discussion section.

Round 2

Reviewer 1 Report

Comments and Suggestions for Authors

The manuscript significantly improved now. Although I still think that the general idea of diving deeper into the matter should be applied to the other pathways/targets than the ones that have been extensively worked on, one of course can publish the review in this current form.